# An Efficient Way to Screen Inhibitors of Energy-Coupling Factor (ECF) Transporters in a Bacterial Uptake Assay

**DOI:** 10.3390/ijms23052637

**Published:** 2022-02-27

**Authors:** Spyridon Bousis, Steffen Winkler, Jörg Haupenthal, Francesco Fulco, Eleonora Diamanti, Anna K. H. Hirsch

**Affiliations:** 1Helmholtz Centre for Infection Research (HZI), Helmholtz Institute for Pharmaceutical Research (HIPS), Campus Building E 8.1, D-66123 Saarbrücken, Germany; spyrosbousisorg@gmail.com (S.B.); steffen.winkler@hotmail.de (S.W.); joerg.haupenthal@helmholtz-hips.de (J.H.); frfulco@gmail.com (F.F.); eleonora.diamanti@helmholtz-hips.de (E.D.); 2Stratingh Institute for Chemistry, University of Groningen, Nijenborgh 7, 9747 AG Groningen, The Netherlands; 3Department of Pharmacy, Saarland University, Campus Building E8.1, 66123 Saarbrücken, Germany

**Keywords:** energy-coupling factor transporters, antimicrobials, B-type vitamins, bacterial uptake assay, screening

## Abstract

Herein, we report a novel whole-cell screening assay using *Lactobacillus casei* as a model microorganism to identify inhibitors of energy-coupling factor (ECF) transporters. This promising and underexplored target may have important pharmacological potential through modulation of vitamin homeostasis in bacteria and, importantly, it is absent in humans. The assay represents an alternative, cost-effective and fast solution to demonstrate the direct involvement of these membrane transporters in a native biological environment rather than using a low-throughput in vitro assay employing reconstituted proteins in a membrane bilayer system. Based on this new whole-cell screening approach, we demonstrated the optimization of a weak hit compound (**2**) into a small molecule (**3**) with improved in vitro and whole-cell activities. This study opens the possibility to quickly identify novel inhibitors of ECF transporters and optimize them based on structure–activity relationships.

## 1. Introduction

Energy-coupling factor (ECF) transporters are a class of transmembrane proteins that facilitate the uptake of micronutrients such as B-type vitamins and cations inside the cell. They belong to the superfamily of ATP-binding cassette (ABC) transporters and actively import several micronutrients by consumption of ATP. Unlike eukaryotic ABC transporters that facilitate both uptake and extrusion of compounds from organelles, the ECF transporters are classified and reported only as importers of micronutrients [1]. The architecture of the ECF transporters consists of a complex of four proteins: one transmembrane protein (T-component), two cytosolic ATPases (A-components) forming together the energy-coupling module (EcfAA’T), and a substrate-specific binding protein (S-component) [2]. Based on the genomic location of the S-components, they are classified into groups I and II (Figure 1).

Specifically, in group I, a “dedicated” EcfAA’T interacts with a single S-component and in this class, both EcfAA’T and S-component are encoded in the same operon. For instance, both CbiMNQO and NikMNQO belong to group I of ECF transporters that mediate cobalt and nickel ion uptake, respectively [3]. Regarding group II, a single “shared” EcfAA’T interacts with various S-components and is responsible for the import of various micronutrients. In this case, the module and the substrate-specific protein are not encoded in the same operon, but copies of different S-components are scattered across the chromosome. ECF FolT and ECF ThiT are two examples of this subgroup (II) that transport folic acid and thiamine, respectively [4]. The importance of the ECF transporters resides in their taxonomic distribution; they are present only in prokaryotes and, in contrast to other ABC transporters, they are absent from eukaryotes, making them promising targets for the treatment of bacterial infections. A large number of bacteria, including pathogens such as Enterococcaceae, Streptococcaceae, and Clostridia utilize ECF transporters to maintain the homeostasis of several vitamins [5]. For example, *Streptococcus pneumoniae* uses ECF transporters to salvage a variety of B-type vitamins such as pantothenate, riboflavin, niacin, pyridoxal, and biotin. These pathogens cause a significant nosocomial and economic burden [6]. In addition, some of them, such as vancomycin-resistant *Enterococcus faecium* and penicillin-non-susceptible *S. pneumoniae*, are classified by the World Health Organization as high and medium priority pathogens for the development of new antibiotics, respectively [7]. Furthermore, ECF transporters are also present in non-pathogenic organisms such as Lactobacillaceae. Specifically, *Lactobacillus lactis* and *Lactobacillus delbrueckii* have been extensively used as model microorganisms, providing valuable insights into the transport mechanism as well as the physical 3D conformation of the ECF transporters [2]. Notably, *L. lactis* is one of the first organisms that was used to study the transport of folic acid into the bacterium mediated by an ECF transporter [8]. Although ECF transporters have been recently reported as potential antibiotic targets, they remain underexplored from a medicinal point of view. In order to identify and evaluate novel inhibitors by cell-based screening the use of an assay that allows for a certain throughput is necessary. ECF transporters are transmembrane proteins; therefore, their purification is challenging, while the stability of both the whole complex and the ECF module is trivial. In our first report, we used an in vitro transport–activity assay using purified ECF FolT2 reconstituted in proteoliposomes to identify hits **1** and **2** (Figure 2) as inhibitors of the ECF FolT2 transporter [9]. Specifically, we used the recombinant *Escherichia coli* MC1061 strain that cannot endogenously express ECF transporters but rather heterogeneously to express and purify the folic acid transporter. Then, the ECF FolT2 was successfully reconstituted into proteoliposomes using it for an in vitro evaluation identifying the first inhibitors of the ECF transporters among a list of compounds. Following our previous study, here we present the establishment of a convenient functional, whole-cell-based transport assay using *L. casei* in 96-well plates.

The new whole-cell uptake assay was used to further validate and develop hit **2** into a more potent compound through a structure–activity relationship (SAR) study improving its properties and antibacterial activity. This achievement was accomplished by two complementary approaches: biochemical evaluation of a commercial library of structurally related compounds of hit **2** and a classical SAR study.

To demonstrate the importance and the throughput of our novel plated whole-cell uptake assay, we performed a parallel study in which we conducted an SAR study, leading to the identification of low-micromolar inhibitors of the ECF FolT2. [10] Specifically, our in-house library was screened virtually followed by in vitro evaluation of twenty selected hits using the new assay. To expand the SAR, we synthesized and evaluated a library of derivatives, leading to a new chemical class of potent substituted ureidothiophene inhibitors with an IC_50_ value of 2.1 µM on target and a single-digit micromolar MIC value against several Gram-positive strains including *S. pneumoniae*.

## 2. Results

Inspired by the way Henderson [8] measured the transport of various vitamins mediated by ECF transporters, we developed a functional bacterial uptake assay to evaluate the ability of molecules to inhibit the transport of vitamins by an ECF group II transporter.

In our previous study, we showed that bacterial cells can be used efficiently to biologically evaluate compounds as potential inhibitors of an ECF transporter [9]. Specifically, an *Escherichia coli* strain that expresses heterologous ECF FolT2 from *L. delbrueckii* was used to demonstrate on-target activity. In the present study, we selected a Gram-positive strain (*L. casei*) due to its ability to import several essential vitamins such as folic acid, riboflavin, thiamine and biotin from the environment. This strain encodes all the necessary genes for the expression of EcfAA’T as well as the S-components (FolT, RibU, ThiT, and BoiY) that interact with and bind to the respective vitamins [5].

Although *L. casei* is not pathogenic, it constitutes an excellent choice as it is one of the best-studied model microorganisms for the transport of folic acid and other vitamins mediated by ECF transporters. Furthermore, it has no additional barriers (outer membrane) that the compounds need to overcome to enter the bacteria. Since *L. casei* is auxotrophic for folic acid, it was chosen over other strains that regulate the homeostasis of folate by both uptake and *de novo* biosynthesis. It is considered that the uptake would be higher in an auxotrophic strain. In an attempt to imitate the physiological conditions, we developed a transport assay in *L. casei* instead of *E. coli*, as the ECF transporters are naturally present in Gram-positive but not in Gram-negative strains. A bacterial uptake assay has many advantages over an in vitro assay (Table 1).

Importantly, the speed of testing increased dramatically as this assay can be performed in 96-well format. The challenging step of purifying a transmembrane protein can be avoided, reducing the amount of work. In addition, the bacterial uptake assay is a multiparameter assay as the efficacy and the permeability across biological barriers are considered. Finally, the allosteric mode of inhibition can be addressed by changing the substrate (vitamin imported by different S-components), while in an in vitro setup, a second protein needs to be purified and reconstituted in liposomes. Consequently, inhibition of the uptake of different vitamins by the desired compounds points to a targeted, allosteric inhibition of the energy-coupling module rather than inhibiting the S-component, leading to reduced uptake of all ECF transporter-mediated vitamins and a multifaceted disturbance of metabolic cellular activity and growth.

## 3. Discussion

The development of the bacterial uptake assay required recurring iterative optimization of the assay conditions. Initially, we observed a strongly varying uptake rate in different cultures. Because cellular activity strongly varies during different growth phases, including enzyme expression levels, the availability of nutrients, or viabilities, the assay was preceded with precultivation of *L. casei* reaching the beginning of the exponential growth phase (OD_600_ = 0.5). Furthermore, possible interfering substances, such as residual vitamins in the medium, were removed prior to the assay by resuspending the cells in citrate buffer including 2% D-glucose. Glucose had been reported to enhance the uptake rate of folic acid in *L. casei* by increasing ATP availability [11]. This is reasonable since it is the co-substrate in ATPases and the driving force of the ECF transporter. Furthermore, 96-well filter plates in combination with a vacuum manifold were used, which markedly accelerated washing steps compared to centrifuge-based separation techniques ensuring a rapid screening of test compounds (Figure 3).

To further investigate folic acid uptake in *L. casei*, Michaelis–Menten kinetics were analyzed reporting a *K_M_* of 49 (±4) nM (Figure 4) that is similar to other published values. Both compounds **1** and **2** were designed to inhibit the EcfT domain allosterically—altering the maximal uptake rate, but not the *K_M_*. Thus, the folic acid concentration could be adjusted to a level that induces uptake rates close to the maximum uptake rate (5–10 times higher than *K_M_*), where the effects of allosteric inhibitors are supposed to be more visible and competitive inhibitors are identified as less active compounds. However, higher folate concentrations would have either reduced the fraction of radiolabeled folate or increased the total amount of radiolabeled folate. In turn, this resulted either in a lower signal and a lower accuracy of the assay or in markedly higher costs, so that the folate working concentration was set to 50 nM.

Furthermore, the allosteric mode can be further investigated by using other vitamins, such as riboflavin, thiamine, or biotin, as discussed above.

Finally, the IC_50_ of **1** was determined at 282 ± 108 µM for the uptake into proteoliposomes assay [9] and it exhibited an IC_50_ value of 315 ± 15 µM in the bacterial uptake assay showing a comparable potency between the two assays (Appendix A). Thus, hit **1** was chosen as a positive control for all following assays. We also showed that **HIPS5031** a close derivative of **1**, is able, at a concentration of 69.66 μM, to inhibit the folate uptake of 41.8 ± 10.3% [12].

To ensure the compounds have time to enter the bacteria and steady-state binding of the target, the compounds were mixed with the bacterial suspension and incubated prior to starting the transport reaction by folate addition. Impairments that may be caused by DMSO as an organic solvent for the test compounds, such as decreased viability or further unforeseen side effects, were not observed for DMSO working concentrations of up to 10%, allowing the screening of more hydrophobic compounds (Appendix A).

Having established the transport assay using **1** as a reference compound, firstly, we measured the inhibition of the uptake of folic acid by **2** (Figure 5). Whereas **2** exhibits an IC_50_ value of 1.2 mM in the in vitro assay in *L. delbrueckii*, it is inactive up to 1.2 mM in the bacterial uptake assay *L. casei* (Table 2) [9]. We reasoned that the difference in activity may be ascribed either to permeability issues of the compound or to a difference in the amino acid sequence of the ECF transporter between *L. delbrueckii* and *L. casei*. In order to observe the differences in the sequence of the ECF FolT transporters between the two strains, we conducted a sequence-conservation study. For the alignment of the four domains of the ECF transporter from both strains (FolT, EcfT, EcfA1, and EcfA2) we used FASTA files we retrieved from the UniProtKB database [13] (Appendix A). Although in general, the sequence identity between the two strains is not high (Appendix A), as we depicted in a previous study, regions of druggable pockets show very high sequence-identity conservation [5]. Even if both assays rely on ECF transporters, differences both in the sequence and in the nature of the assay may cause variations in the activity.

### Structure–Activity Relationship (SAR) Study

In order to explore the chemical space around compound **2**, we extended our screening to a small library of structurally related compounds, which were previously obtained through commercial sources or in-house chemical synthesis. Most of the compounds were tested at 500 μM, and less soluble ones were tested at their maximum soluble concentration. From the nineteen compounds (**4**–**22**) tested, seven (**4**, **5**, **10**, **11**, and **16**–**18**) showed more than 20% inhibition, while the others did not show any ability to influence the transportation of the ECF transporters (Appendix A).

In parallel, to address the permeability issue caused by the zwitterionic nature of **2**, we rationally designed and synthesized molecules with broad modifications and devoid of zwitterionic nature. To do so, we substituted the secondary aliphatic amine with an aniline maintaining the carboxylic acid. Interestingly derivative **3**, which bears the COOH group in *m*-position, is seven-fold more potent compared to the initial hit **2** (Table 3).

Then, we tested the sensitivity of the ortho- and para-position of the COOH group within the distal phenyl ring (**23** and **24**). While compound **23** turned out to be insoluble under the conditions of the assay, **24** showed no inhibition up to 500 μM. Next, we explored the left-hand ring that bears the chloride atom and observed that changing the substitution pattern of the chloride from para- to meta- to ortho- does not influence the cell-based activity (see **3**, **25**, and **26**, respectively Table 3). This finding makes the left-hand ring a good place for potential fragment growth. In a third step, we designed molecules having modifications in the middle furan ring. Therefore, we shifted the substitution pattern from _1_C–_4_C to _1_C–_3_C (**27**), which surprisingly abolished the activity of the molecule (Table 4).

To examine the antimicrobial activity of **3** as the best compound in this study, a panel of Gram-positive pathogens that rely on the ECF have been screened. Encouragingly, **3** showed good antimicrobial activity against both *S. aureus* and *S. pneumoniae* strains (Table 5) [4].

## 4. Materials and Methods

### ECF-T Inhibition Assay

Ten milliliters of MRS medium were inoculated with 50 µL of a −80 °C cryo-culture of *L. casei* (strain: (Orla-Jensen 1916) Hansen and Lessel 1971 (DSM-#. 20011)) and grown at 30 °C for 17 h at 180 rpm in an Erlenmeyer flask. The OD600 was measured in a Nanodrop 2000c spectrophotometer (Thermo Fisher Scientific, Waltham, MA, USA) using half-micro cuvettes. The culture was diluted to OD_600_ = 0.06 in 25 mL and grown under the same conditions as before for 16 h. The new overnight culture was diluted to OD_600_ = 0.4 in 25 mL MRS medium and regrown in a 50 mL reaction tube for 2 h under the same conditions. The culture was subsequently centrifuged, washed with 50 mL PBS (50 mM, pH 7.4), centrifuged again, and resuspended in citrate buffer (20 mM, pH 6, 2% D-Glucose, 8 g/L NaCl, 0.2 g/L KCl) while adjusting the OD_600_ to 0.5. Then, 185 µL of this cell suspension was subsequently distributed on an opaque MultiScreen HTS Filter Plate—GV (Merck KGaA, Darmstadt, Germany). For each plate, two wells containing buffer for blank determination were included. A stock solution of compounds in DMSO (10 µL) was added to the wells and mixed (final DMSO conc. = 5%), followed by a preincubation at RT for 10 min. The solubility of the compounds in citrate buffer was verified in 96-well plates. Poorly soluble compounds were tested at the highest possible concentration. In addition, for each plate, four negative controls containing DMSO instead of inhibitor and a positive control were included. Tritium-labeled folic acid (Moravek Biochemicals, Brea, CA, USA) (5 µL, 2 µM, radioactivity was adjusted to 12.7 Ci/mmol using unlabeled folic acid) was added to the wells leading to a final concentration of 50 nM. The solutions were mixed (final volume 200 µL) as before, and the plate was incubated in a preheated incubator at 30 °C for 30 min. The plate was covered with sealing tape during the incubation steps to avoid evaporation. To stop the reaction, the liquid in the filter plates was removed from the cells by centrifugation (3 min at 1500× *g*). Subsequently, the cells were washed twice with PBS (10 mM, pH 7.2), followed by centrifugation steps (3 and 5 min at 1500× *g*). After removal of the underdrain, residual liquid on the bottom of the filters was removed using lint-free paper towels and the plate was dried at 50 °C in the incubator. After 10 min, 30 µL of liquid scintillating fluid was added into each well, the plate was covered with sealing tape and gently vortexed for 5 min. Relative radioactivity was determined for 1 min each well in a Wallac MicroBeta TriLux BetaCounter (Perkin Elmer, Waltham, MA, USA). The data were analyzed by using OriginPro 2018G (OriginLab Corporation, Northampton, MA, USA). Counts were averaged and after subtraction of the blank, the percentage of inhibition was determined by the following formula:$$Inh.\;\%=100-\frac{{\bar{x}}_{\mathrm{inhibitor}}}{{\bar{x}}_{\mathrm{negative}\;\mathrm{control}}}\times 100$$

The data were plotted including standard deviations. IC_50_ values were determined by plotting percentages of inhibition against the related concentrations and fitting the Origin standard function “Hill1” to the data. The *K_M_* was determined by plotting the relative radioactivity against the folic acid concentration and fitting the Origin standard function “Hyperbl” to the data.

## 5. Conclusions

We established a working platform based on 96-well plates that allows for a quick and convenient identification of ECF transporter inhibitors. This protocol is based on a bacterial uptake assay using *L. casei* as a model microorganism. The assay has the advantage of identifying compounds that inhibit ECF transporters in a whole-cell environment. We demonstrated the efficiency of this approach by the successful optimization of hit **2** into compound **3** (Figure 5) endowed with an improved activity. In addition, compound **3** exhibits good antimicrobial activity against *S. pneumoniae* strains. In sum, our results show that it is possible to develop small-molecule inhibitors that block the entrance of vitamins into bacteria through inhibition of ECF transporters and thereby the growth of pathogenic bacteria. Our bacterial uptake assay is amenable for screening of compound libraries and enables the optimization of inhibitors of ECF transporters. In addition, this assay can be extended to other members of the ECF transporter family by changing the natural substrate used for the uptake, providing a versatile screening platform.

## Figures and Tables

**Figure 1 ijms-23-02637-f001:**
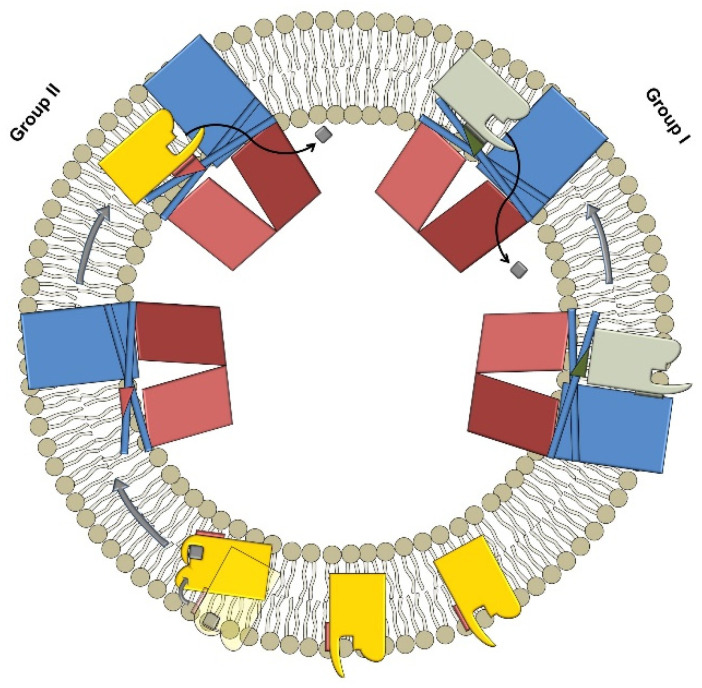
Schematic representation of group I and II ECF transporters. The heterodimeric ATPases (ECFA and ECFA’) are represented in light and dark red color, while the T-component is shown in blue. The S-components belonging to the ECF group I and II are colored in light green and yellow, respectively.

**Figure 2 ijms-23-02637-f002:**
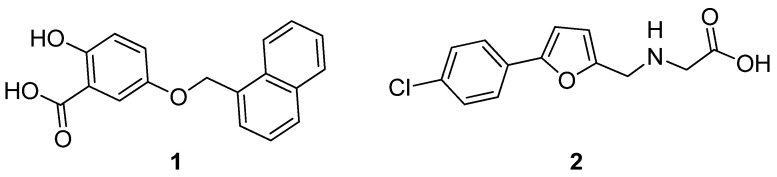
Chemical structures of the two identified inhibitors of the group II ECF-FolT2 transporter [9].

**Figure 3 ijms-23-02637-f003:**
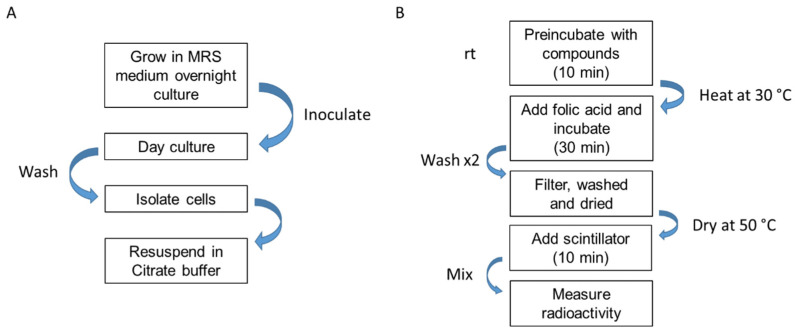
Illustration of the key steps of the new plated whole-cell uptake assay of ECF FolT transporter. (**A**) Depicting the cell-preparation steps of *L. casei*. Wash: PBS (50 mM, pH 7.4). MRS: De Man, Rogosa and Sharpe agar. (**B**) Depicting the crucial steps during the assay. Wash: PBS (10 mM, pH 7.2) citrate buffer (20 mM, pH 6, 2% D-Glucose, 8 g/L NaCl, 0.2 g/L KCl) and 5% DMSO.

**Figure 4 ijms-23-02637-f004:**
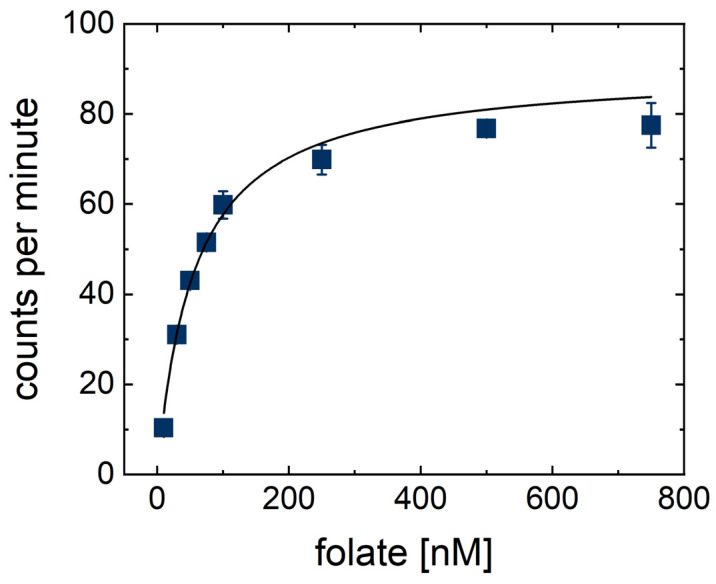
Substrate-dependent folate uptake in *L. casei* in PBS (50 mM, pH 7.4, 2% Glucose). With *K_M_* = 49 (±4) nM and Vmax = 83 (±2) counts per minute (cpm).

**Figure 5 ijms-23-02637-f005:**
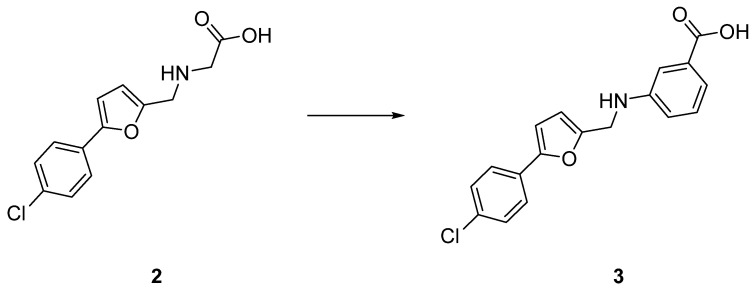
Chemical evolution of **2** to **3** presented on this work.

**Table 1 ijms-23-02637-t001:** Advantages of a bacterial uptake assay over uptake into proteoliposomes. The bacterial uptake assay is performed in *L. casei*, whereas the uptake into proteoliposomes is based on the isolated ECF FolT2 transporter from *L. delbrueckii*.

Bacterial Uptake Assay	Uptake into Proteoliposomes
Medium to high throughput	Low throughput
No protein purification	Requires protein purification
Multiparameter optimization	Single-parameter optimization
Monitor the uptake of several vitamins	Monitor the uptake of specific vitamin

**Table 2 ijms-23-02637-t002:** IC_50_ inhibition values in different assays.

Name	IC_50_	Assay Type of ECF-FolT
**2**	>1.2 mM	Proteoliposome uptake
**3**	204 ± 22 µM	Bacterial uptake

**Table 3 ijms-23-02637-t003:**
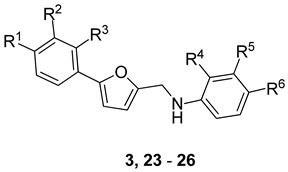
SAR of the aniline derivatives. Biological evaluation of **3** and **23**–**26** for inhibition of the ECF FolT transporter-mediated folate import in the *L. casei* based uptake assay.

Name	R^1^	R^2^	R^3^	R^4^	R^5^	R^6^	Inhibition at 200 μM
**3**	Cl	H	H	H	COOH	H	43% (IC_50_ = 204 ± 22 μM)
**23** ^1^	Cl	H	H	COOH	H	H	11% at 50 µM
**24**	Cl	H	H	H	H	COOH	n.i at 500 μM
**25**	H	Cl	H	H	COOH	H	41%
**26**	H	H	Cl	H	COOH	H	36%

^1^ Not soluble above 50 μM. n.i. = no inhibition (if inhibition < 10%).

**Table 4 ijms-23-02637-t004:**
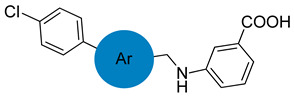
SAR of the derivatives that bear modifications at the middle ring. Biological evaluation of **3** and **27** for inhibition of the ECF FolT transporter-mediated folate import in the *L. casei*-based uptake assay.

Name	Ar	Inhibition at 200 μM
**3**	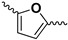	43%
**27**	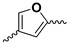	8%

**Table 5 ijms-23-02637-t005:** Antibacterial profile of **3**.

Indicator Strain	Auxotrophic	MIC (μg/mL)
*Staphylococcus aureus* str. Newman	no	23
*Streptococcus pneumoniae* DSM-20566	yes	21
*Streptococcus pneumoniae* DSM-11865 ^1^	yes	10

The auxotrophic status indicates which of the strains is auxotrophic for at least one B-vitamin. ^1^ PRSP: penicillin-resistant *Streptococcus pneumoniae*.

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
