# Peer review of "An Efficient Way to Screen Inhibitors of Energy-Coupling Factor (ECF) Transporters in a Bacterial Uptake Assay"

_ijms, 2022, doi:10.3390/ijms23052637_

Round 1

Reviewer 1 Report

Dear Editor and Authors

The manuscript Bousis et al with the Title:” An efficient way to screen inhibitors of energy-coupling factor 2 (ECF) transporters in a bacterial uptake assay” is well designed, prepared and written. ABC- Transporter are very essential transporter systems in prokaryotic as well as in eukaryotic cells. While the human ABC transporters act exclusively as efflux transporters, the bacterial ABC transporters are important for the uptake of nutrient and vitamins. Therefore, they are important targets for the development of therapeutic drugs.

Nevertheless, the following questions and suggestions should be addressed.  

  1. Does Group II ECF-Folto2 transporter perform Efflux of its Substrate comparable to the Human ABC-Transporter or they mediate only uptake
  2. Four genes of the ECF FolT2 from L. delbrueckii are expressed in Escherichia coli. Does Escherichia coli express comparable endogenous ECF FolT2 transporter.
  3. Why did you not performed a concentration dependent inhibition by the bacterial uptake assay with casei comparable to the uptake into proteoliposome assay for L. delbrueckii  
  4. You mentioned in line 163 and 164 that the difference in activity may be to a difference in the amino acid sequence of the ECF transporter of between delbrueckii and L. casei. Have you compared the sequence-by-sequence alignment? How many percent is the amino acid identity or homology of the ECF transporter between L. delbrueckii and L. casei.  
  5. You stated in line 175 and 176 for nineteen compounds more than 20% inhibition, while the others did not show any ability to influence the transportation of the ECF transporters (Figure S12), unfortunately I could not find the Figure S12 in the  supplementary documents
  6. The transport rate were plotted as a relative radioactivity to determine the Km value and the Vmax value are presented unfortunately as relative value, which is without precise dimension e.g. pmol/min/mg (protein). Pleas calculate the uptake rate and Vmax in pmol or nmole. Since you have the specific activity and the concentrations of radioactive labeled folic acid.
  7. Can you remove the inhibition in the following sentence in line 150 and 151 and change to “We also showed that HIPS5031 a close derivative of 1, is able @ 69.66 µM concentration to inhibit the folate uptake of 41.8±10.3%”.
  8. In line, 180 add hour (17 h)

Reviewer 2 Report

The article "An efficient way to screen inhibitors of energy-coupling factor 2 (ECF) transporters in a bacterial uptake assay" by Bousis et al describes a new way to screen some inhibitors.

I find some problems with the way data are presented, as they are right now it does not look like the work behind this manuscript is a lot, so I think that the whole presentation should be improved.

The authors talk about a new protocol, but if this is the key point , I suggest adding a figure in which the key steps are summarized. The authors claim that this test could be done in a 96. well plate and that the test is high throuput, but the molecules tested are only few. Moreover, not even one kinetic is shown. It would be mice to see the raw data from which table 2 was derived.

The layout of that paper is also strange: no discussion is mentioned, although I think it is present to some extent. point 2 mentioned "Results" and point 3 is titled:"Structure-activity relationship (SAR) study", but aren't these also results? 4. is for materials and methods, and if only one method is present, the paragraph should not be 4.1 if there is no 4.2....Introduction and discussion should be extended, only 10 references?I suppose that the impact of a new protocol should involve a broader audience.

In addition, 2 fig.3 are present. Strains are not always written in italic

Round 2

Reviewer 2 Report

No more comments, thanks for taking into account my suggestions.